# Estimated transmission dynamics of SARS-CoV-2 variants from wastewater are unbiased and robust to differential shedding

David Dreifuss [1,2], Jana S. Huisman [1,2,3] ✉, Johannes C. Rusch[4], Lea Caduff [4], Pravin Ganesanandamoorthy[4], Alexander J. Devaux[4], Charles Gan [4], Tanja Stadler [1,2], Tamar Kohn [5], Christoph Ort [4], Niko Beerenwinkel [1,2] ✉ & Timothy R. Julian [4,6,7] ✉

The COVID-19 pandemic has accelerated the development and adoption of wastewater-based epidemiology. Wastewater samples can provide genomic information for detecting and assessing the spread of SARS-CoV-2 variants in communities and for estimating important epidemiological parameters such as the selection advantage of a viral variant. However, despite demonstrated successes, epidemiological data derived from wastewater suffers from potential biases. Of particular concern are shedding profiles, which can affect the relationship between true viral incidence and viral loads in wastewater. Changes in shedding between variants may decouple the established relationship between wastewater loads and clinical test data. Using mathematical modeling, simulations, and Swiss surveillance data, we demonstrate that estimates of the selection advantage of a variant are not biased by shedding profiles. We show that they are robust to differences in shedding between variants under a wide range of assumptions, and identify specific conditions under which this robustness may break down. Additionally, we demonstrate that differences in shedding only briefly affect estimates of the effective reproduction number. Thus, estimates of selective advantage and reproduction numbers derived from wastewater maintain their advantages over traditional clinical data, even when there are differences in shedding among variants.

In the context of the COVID-19 pandemic, wastewater-based surveillance has become widespread and has proved to be a reliable data source to estimate disease trajectories[1]. The viral loads of SARS-CoV-2 in wastewater have a strong positive correlation with COVID-19 incidence in the area connected to the studied sewershed, and they are routinely used as an epidemiological indicator. Tracking the dynamics of SARS-CoV-2 RNA in wastewater can reliably estimate the effective reproductive number ($R_e$), a critical parameter describing disease dynamics and informing public health policy[2]. Further, wastewater samples enable the detection and tracking of the progression of genomic variants using PCR-based methods[3], including quantitative and digital PCR (dPCR)[4], as well as next-generation sequencing (NGS)

[1]Department of Biosystems Science and Engineering, ETH Zurich, Basel, Switzerland. [2]SIB Swiss Institute of Bioinformatics, Lausanne, Switzerland. [3]Physics of Living Systems, Massachusetts Institute of Technology, Cambridge, MA, USA. [4]Eawag, Swiss Federal Institute of Aquatic Science and Technology, Dübendorf, Switzerland. [5]Laboratory of Environmental Virology, School of Architecture, Civil and Environmental Engineering, École Polytechnique Fédérale de Lausanne (EPFL), Lausanne, Switzerland. [6]Swiss Tropical and Public Health Institute, Allschwil, Switzerland. [7]University of Basel, Basel, Switzerland. ✉e-mail: jhuisman@mit.edu; niko.beerenwinkel@bsse.ethz.ch; tim.julian@eawag.ch

methods[5–7]. Both approaches are able to reliably quantify the relative abundances of genomic variants, with estimates coinciding with those obtained from traditional methods based on clinical samples[4–7]. Wastewater genomics allows early detection of variants, as shown, for example, by the detection of the first confirmed case of Omicron BA.1 in Switzerland in wastewater[8]. For epidemiology, wastewater samples provide accurate population-level data at largely reduced costs and simplified logistics. Therefore, wastewater-derived data is increasingly incorporated into national surveillance strategies[9,10].

Tracking the relative abundance of a newly introduced genomic variant over time provides insights into its dynamics and selection advantage over the current dominant strain, which is a key parameter for models predicting future disease trajectories. Early, accurate estimates of this parameter are crucial as they may highlight replacement and can inform on the epidemic potential of a new variant before an increase in case numbers is observed[11]. The growth or selection advantage informs about the increase in transmissibility and immune evasion of a variant[12]. Generally, it is assumed that a constant multiplicative selection advantage holds for a newly increasing variant, which leads to logistic growth of its relative abundance[13]. It has been shown that both NGS and dPCR analysis of wastewater samples can provide timely and accurate estimates of the selection advantage of a variant, requiring orders of magnitude fewer samples than traditional surveillance methods relying on samples of infected individuals[4,5].

However, wastewater-based quantification of SARS-CoV-2 and its genomic variants is affected by shedding load profiles, which describe the average amount of viral RNA each infected individual contributes to the sewer during the course of infection[2]. Although some rare prolonged low levels of fecal shedding have been reported[14], shedding during the later course of infection is believed to decay exponentially with a short half-life (1–2 days), such that wastewater loads are indicative of COVID-19 incidence[15]. The different variants of concern that have emerged and spread since the beginning of the pandemic have all exhibited a difference in average viral loads in the respiratory tract, but the overall shape of the shedding profile was conserved[16]. This pattern has been similarly observed in fecal shedding, where variations in total shedding between variants are reported, but the overall profile shape remains consistent[17]. For the Omicron BA.1 variant of concern, lower fecal shedding compared to Delta[18] has been suspected, possibly due to changes in tropism[19]. Such differences in shedding profiles can decouple the established relationship between estimates of incidence based on wastewater from those derived from clinical test data. As a consequence, population-level incidence based on historical relationships between SARS-CoV-2 concentrations in wastewater and case data will be underestimated (if shedding from the new variant is lower) or overestimated (if shedding from the new variant is higher). To enable accurate wastewater-based estimates of disease trajectory for all variants, including new and uncharacterized ones, analytical approaches need to be robust to shifts in shedding load profiles. Such methods will aid perennial integration of wastewater-derived epidemiological data into public health policy.

In the present study, we explore the bias that shedding may introduce to estimates of the selective advantage of a variant, and the robustness of such estimates to changes in shedding between variants. We derive closed-form expressions of the bias in the wastewater-based estimators of selection and perform simulations. Our derivations and simulations demonstrate that under common assumptions for models of selection, the wastewater-based estimates are both unbiased with respect to shedding and robust to arbitrary changes in shedding. We use these derivations and simulations to show that, when variants have different total shedding, estimates are robust even when relaxing the main assumptions of the selection models (i.e., constant transmission, constant selection, and constant mean generation times). We compare

estimates based on matched wastewater and clinical data in Switzerland obtained during the introduction of BA.1, a variant with apparent differences in shedding from the circulating Delta variant, to demonstrate the robustness of wastewater-based estimates of selection on real data. Additionally, we show that differences in shedding briefly impact estimates of the overall effective reproduction number, and we provide approximate bounds on the maximal bias.

## Results

When using wastewater-derived data, epidemiological analysis is complicated by the shedding load profile of the virus, which is defined as the average shedding of viral particles through time after infection. This profile can cause the observed viral loads $X^w(t)$ in wastewater to differ from the true incidence $X(t)$. The total amount $l$ of viral particles shed throughout an infection will scale $X^w(t)$ with respect to $X(t)$, and the mean shedding time $\mu$ introduces a delay (Fig. 1, Supplementary B, Supplementary Fig. 4). As a result, the relationship between observed loads and true incidence can be approximated as

$$X^w(t) \approx lX(t - \mu) \qquad (1)$$

Here, we determine when the estimates of selection advantage of a variant relative to another variant can be estimated in an unbiased and robust manner using measurements of the relative loads of those variants in wastewater (Supplementary Table 1, Fig. 2). We say that the estimate of the selection advantage based on relative loads is unbiased if it does not differ from the estimate based on the true relative incidences. We say that it is robust if a difference in shedding load profiles between the two variants does not affect the estimate.

### Estimates of selection are unbiased and robust under common model assumptions

A common assumption in evolutionary models used to estimate the fitness advantages of SARS-CoV-2 variants is that the transmission rates (the per capita rate at which infectious individuals generate new infections) of both variants remain constant during the period studied. Under that assumption, the shedding load profile only affects the wastewater loads by scaling them with respect to incidence, but does not further distort the signal. (see Methods, Supplementary C). As a result, the observed progression of the relative loads will appear only shifted in time with respect to the true relative incidence. We derive a closed-form expression for the bias in the fitness advantages estimated from relative loads (Supplementary C i). It demonstrates that the bias is zero. Additionally, we illustrate with two common scenarios of interest where this assumption holds: two variants with different fitness are introduced at the same time and compete; one fitter variant is introduced and sweeps the formerly dominant strain (Fig. 2 panels D,H). We simulate the resulting loads in wastewater using shedding load profiles of different total shedding and shapes of the shedding load distributions (Fig. 2 panels A, B, C). These simulations illustrate that the bias in the selection advantage with respect to shedding profiles is zero (Fig. 2 panels E, I). Furthermore, we similarly demonstrate that arbitrary changes in the shape or scale of the shedding load profile have no effect on the estimated fitness advantage, i.e., it is robust (Fig. 2 panels F, G, J, K, Supplementary Equation 18).

### Estimates of selection are always robust to differences in total shedding

Differences in shedding between variants affect the total amount shed mainly in the form of scaling of the shedding load profile, rather than the shape of the shedding load distribution[16,17]. Simultaneously, estimates of incidence based on wastewater loads are most strongly affected by changes in total shedding, rather than by changes in the shape of the shedding load distribution (Supplementary Fig. 4, Eq. 1,

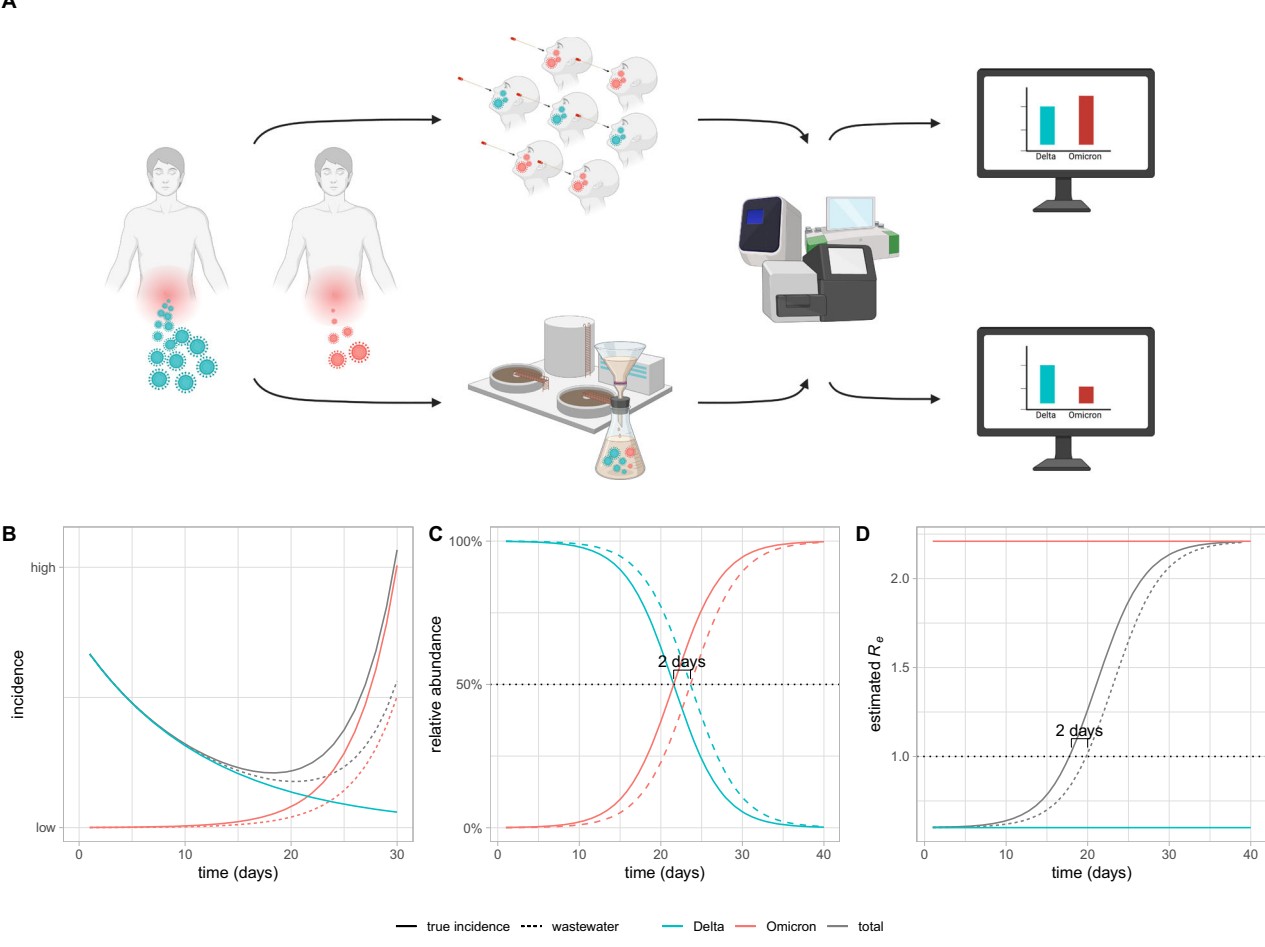

**Fig. 1 | Impact of shedding differences on wastewater-based incidence and $R_e$ estimates, and variant-relative selection advantage. A** Two variants (Delta, blue, and Omicron, red) are shed in different amounts. The incidence of SARS-CoV-2 and its variants in the population (barplots on the right) is quantified from clinical samples (top path) and from wastewater samples (bottom path) using PCR-based and NGS methods. The difference in shedding can lead to differences in incidence estimates. Created in BioRender. Dreifuss, D. (2025) https:// BioRender.com/5azz-vou. **B** Simulated time series: the dominant strain (Delta, blue) has a stable $R_e$ of 0.6, and the incidence is steadily declining. A new variant (Omicron, red) with a selection advantage over Delta resulting in an $R_e$ of 2.2, is introduced and increases in absolute prevalence. We have assumed here that the new variant is shed 50% less, which decouples the historic relationship between community prevalence (solid line) and concentration in wastewater (dashed line), leading to underestimation of its incidence from wastewater-derived data. This in turn also leads to under-estimating the total incidence of the virus (blue). **C** The differential shedding results in a time-delay of 2 days in the growth and decay curves of relative abundance of the variants, but does not alter the growth or decay rates. The estimates of the selection advantage of the variant are not affected. **D** The differential shedding results in a transient bias in the estimated $R_e$. The $R_e = 1.0$ threshold is estimated to be crossed 2 days later than without undershedding. The variant-specific $R_e$ for both variants do not suffer any bias.

Supplementary Section B). To investigate how changes in total shedding might impact the fitness advantages estimated from relative loads, we derive a closed-form expression for the bias, this time including different total shedding (Supplementary Section C ii). We supplement our derivations with simulations, which contain two additional scenarios where the assumption of constant transmission rates is dropped: (i) two variants competing and depleting the number of available susceptibles; (ii) two variants competing and having their transmission rates dropped abruptly by an intervention (Fig. 2 panels L, P). The change in total shedding incurs a delay in the curve of the progression of the relative abundance of the new variant (Fig. 1). In the case where the common assumptions of the selection model hold, the progression of a variant with selection advantage $s$ which is shed differently by a factor $c$ will be delayed by $-\log(c)/s$. However, the estimates of selection advantage are not affected: they are robust to changes in total shedding (Fig. 2, panels F, J, N, R). Furthermore, this robustness holds even when the common assumptions of the selection model are dropped (Fig. 2, panels N, R).

## Estimates of selection are unbiased and robust to changes in generation time

Another common assumption in evolutionary models used to estimate the fitness advantages of SARS-CoV-2 variants is that variants have the same generation time. When this assumption does not hold, estimates of the selection advantage can be biased. Our derivations illustrate that this bias is the same when the selection advantage is estimated from relative loads as it would be using true relative incidences, i.e., the wastewater-based estimates are unbiased (Supplementary C viii). This holds regardless of whether the variants are shed with the same shedding load profile, so we find that the wastewater-based estimates are robust. In particular, this ensures robustness of estimates of selection derived from wastewater in the case where changes in shedding profiles are accompanied by changes in generation time, for example due to an underlying change in the infectivity profile.

In addition we investigate cases where the variants do not only have different generation times, but these also vary in time. This can happen, for instance, due to generation interval contraction[20]. In this

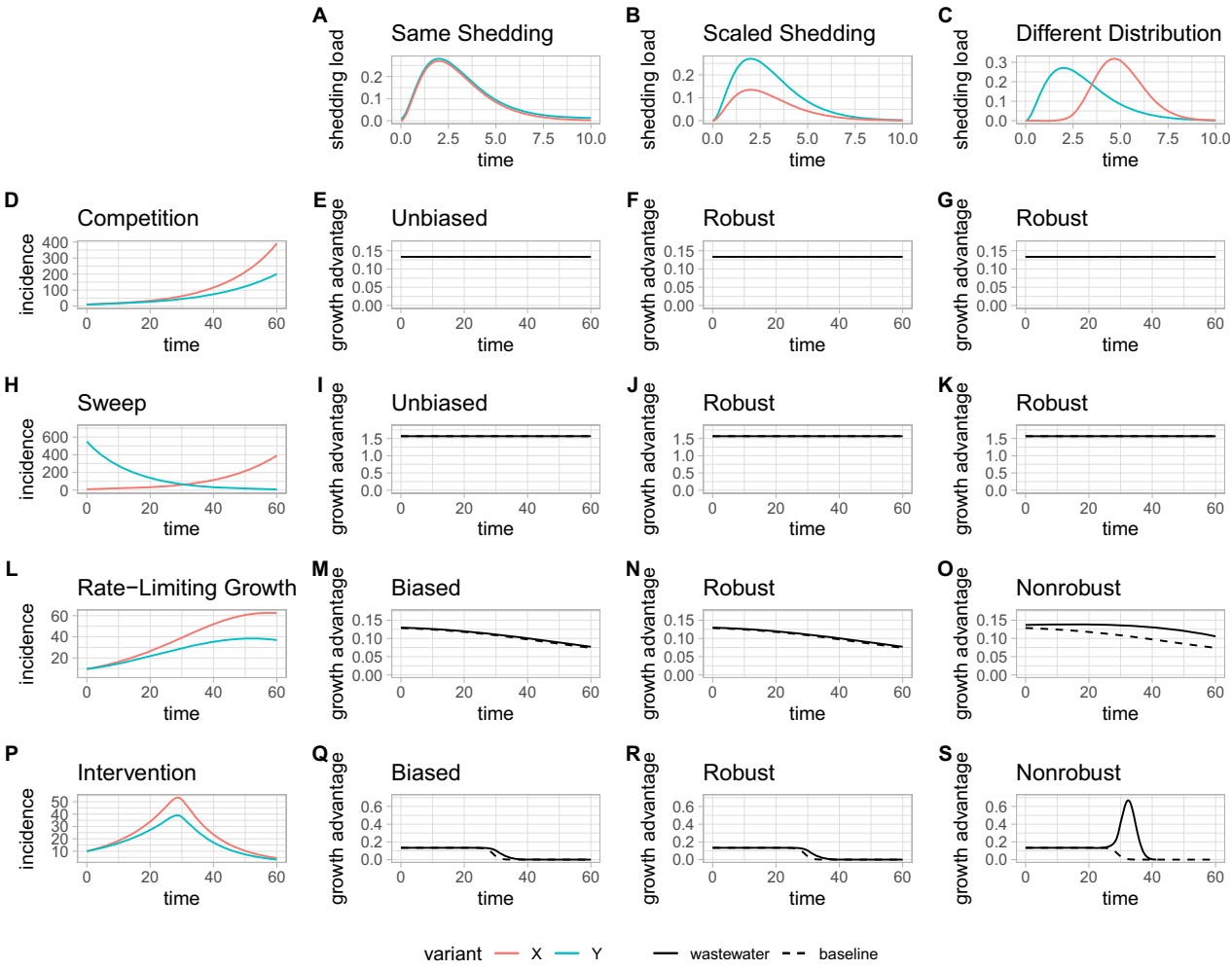

**Fig. 2 | Relative loads-based estimates of selection advantage of $X$ over $Y$ from different time series of incidence of variants $X$ and $Y$ (rows) convolved with different shedding load profiles for variants $X$ and $Y$ (columns), compared to estimates from the ground truth relative incidences (baseline).** We show 3 scenarios for the shedding: same shedding profiles for both variants (**A**), 50% lower total shedding for $X$ relative to $Y$ (**B**), and shedding load distribution for $X$ with different shape relative to $Y$ (but with the same total shedding. **C** Plots at the row index show 4 different scenarios for the dynamics of the incidence of variants $X$ and $Y$: in the first row (**D**), two variants with constant (but different) transmission rates are introduced at the same time; in the second row (**H**), for two variants with constant (but different) transmission rates, one is replacing the other; in the third row (**L**), two variants are introduced at the same time, their transmission rates decreasing smoothly according to the rate-limiting mass action dynamics of a SIR model; in the fourth row (**P**), two variants are introduced at the same time, their transmission rates are abruptly cut to almost zero by an intervention in the middle of the time series. With the standard model assumption of constant transmission rates, the estimates are not biased by the shedding profiles (**E, I**) and robust to changes in total shedding (**F, J**) as well as to changes in shedding dynamics (**G, K**). When this standard assumption is relaxed, the selection advantage does not appear constant anymore even when computed from the ground truth relative incidence, and the apparent selection advantage is delayed in time when estimated from wastewater (**M, Q**), but they are still robust to changes in total shedding (**N, R**). When the shedding dynamics additionally differ between variants, they lose some of their robustness (**O, S**).

case the error introduced by assuming a constant generation time will be similar in the wastewater-based estimates compared to estimates based on true relative incidence data, but delayed in time. The wastewater-based estimates thus remain robust to differences in total shedding even in the presence of different, non-constant generation times.

## Varying apparent selection and arbitrary changes in shedding shape

Models of selection commonly assume that the selection advantage $s$ and transmission rates are constant through time. However, it is possible to imagine scenarios in which the selection $s$ is not constant. More importantly, when dropping the assumption of constant transmission rates, a constant selective advantage does not necessarily appear constant through time anymore (Fig. 2, panels M, Q, Supplementary C

iii.), even when computed from the true relative incidences. In that case, we refer to these instantaneously varying estimates $\hat{s}(t)$ as the apparent selection advantages.

We investigated under which conditions the apparent selection advantage differs when computed from wastewater-derived relative loads. When both variants have the same shedding load distribution, the wastewater-based apparent selection advantage $\hat{s}^w(t)$ will be approximately delayed in time by an amount equal to the mean $\mu$ of the shedding load distribution.

$$\hat{s}^w(t) \approx \hat{s}(t - \mu) \tag{2}$$

As shown above, it is still robust to changes in total shedding between variants (Fig. 2 panels N, R). However, when we additionally have shedding load distributions of different shapes for different

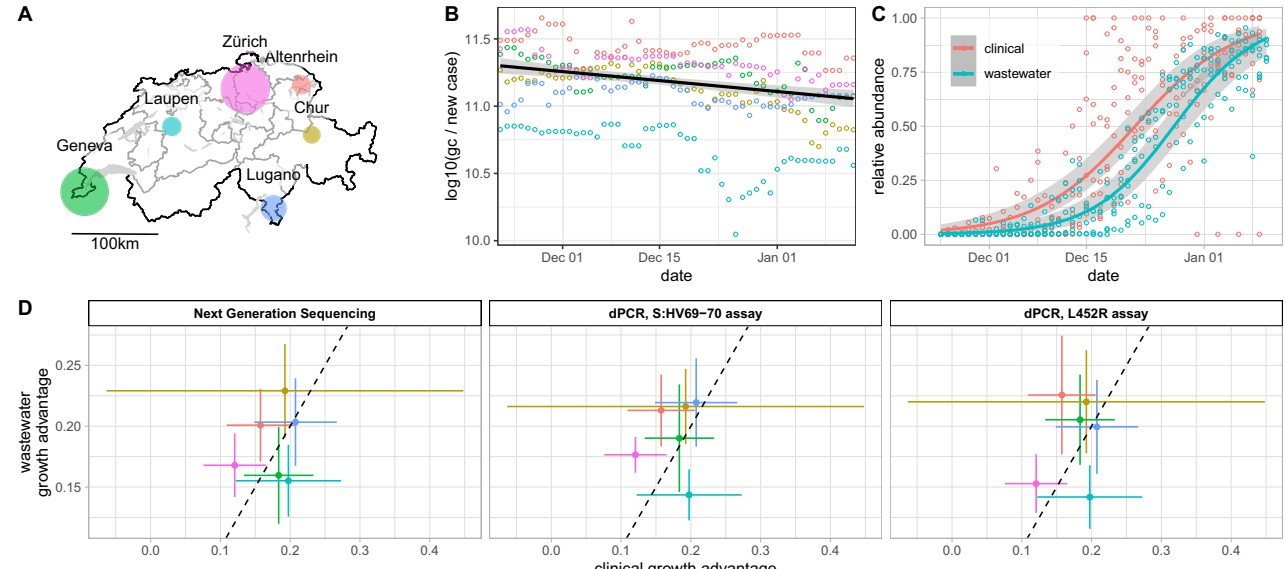

**Fig. 3 | Concordant estimation of the selection advantage of Omicron BA.1 in Switzerland using clinical and wastewater-derived data. A** Map of Switzerland indicating the location of all six wastewater treatment plants (WWTPs) sampled in this study. The discs are scaled to represent the number of inhabitants in the respective sewersheds. **B** Log10 SARS-CoV-2 genome copies (gc) in wastewater (7-day rolling median) per new confirmed case in the sewersheds (7-day rolling median), throughout the spread of BA.1. The point colors represent the WWTP, and the solid line is a robust linear fit along with 95% confidence bands, showing an average reduction of gc / new case of 47% during the study period. **C** Progression of the Omicron BA.1 variant in Switzerland. Points represent average frequencies of the variant in different locations, estimated from clinical sequences ($n = 8525$, red) or wastewater ($n = 280$, blue) NGS samples. Solid lines represent logistic fits along with 95% Wald confidence bands. **D** Selection advantage estimates for BA.1 in the six WWTP regions (colors), based on wastewater NGS ($n = 280$, left) as well as wastewater ddPCR duplex assays targeting the S:HV69-70 deletion ($n = 79$, center) and S:L452R substitution ($n = 74$, right), compared with estimates derived from clinical sequencing from the cantons surrounding the WWTPs. Points represent maximum likelihood estimates, and error bars represent 95% Wald confidence intervals adjusted for overdispersion. The dashed lines represent the 1:1 relationship.

variants, then robustness is lost (Fig. 2 panels O, S). The added bias in this situation is proportional to the difference in mean of the shedding load distributions of the two variants $\mu_x - \mu_y$, and to the rate at which the transmission rates are varying.

### Differences in total shedding and shedding distribution have a transient and limited effect on $R_e$ estimates

The effective reproduction number $R_e$ is a key metric for quantifying epidemiological dynamics, measuring the expected number of secondary infections from a primary case over time. This quantity can be tracked from absolute loads in wastewater data, and we investigated how its estimation is affected by the introduction of variants with differences in shedding. We derived a closed-form expression (Supplementary Section D) to describe the bias in $R_e$ and performed simulations. The $R_e$ estimates are impacted by differences in shedding, although the bias vanishes once the relative abundances of variants no longer vary over time, i.e., the estimator tends to recalibrate itself after a sweep (Fig. 1D, Fig. 2B). Importantly, the rise in $R_e$ due to the introduction and spread of a competitive variant will still be visible in $R_e$ derived from wastewater viral loads, although it may appear delayed. Additionally, for the case where the shedding load profiles of the variants differ in total shedding by a factor $c$, we derived an upper bound for the bias $b$ such that $b \leq \frac{(c-1)s}{4}$. For the case where the shedding load distributions differ, the bound on the bias will additionally be proportional to the differences in their mean $\Delta_\mu = \mu_y - \mu_x$.

### Dispersion of the generation time interval

Both the common model for the selection advantage $s$ and the model for $R_e$ we have used here assume an exponential distribution of the generation time interval, i.e., the time from primary to secondary infections[21]. To test the sensitivity of our results to this assumption, we have simulated stochastic time series of variant transmission with

different degrees of dispersion in the generation time interval distribution and with differences in total shedding. These did not affect our findings on the robustness of the estimation of selection advantage and $R_e$ to changes in total shedding, neither in the case of underdispersion nor overdispersion of the generation interval time distribution (Supplementary Section D, Supplementary Fig. 5).

### Selection advantage estimates of BA.1 in Switzerland from wastewater and from clinical data

In Switzerland, SARS-CoV-2 viral loads in wastewater are monitored since 2020, and variants are monitored in wastewater since 2021 (https://wise.ethz.ch/). During the progression of Omicron, there was an apparent decoupling of the daily measured SARS-CoV-2 load (viral genome copies per day) from the daily reported new cases in all six of the sewersheds monitored at that time (Fig. 3A, B, Supplementary Fig. 1). At the same time, the test positivity rate increased in Switzerland, indicating that this effect was not a reflection of (and was possibly even partially hidden by) testing practices (Supplementary Fig. 6). We suspected that the discrepancy was due to differential shedding of Omicron versus Delta.

Despite this discrepancy, the estimates of the selection advantages obtained from logistic fits on wastewater NGS and variant-specific dPCR analysis of wastewater were very similar to the estimates based on clinical sequencing data (Fig. 3C, Supplementary Fig. 1). As expected, estimates of the midpoint of the curve (the time at which Omicron overcame Delta) were higher when computed from wastewater data compared to those computed using clinical data (Supplementary Fig. 2Z).

We developed a new duplex digital PCR assay to track the S:L452R mutation. This assay, as well as the previously developed assay targeting the S:HV69-70 deletion, enabled us to track BA.1 using these mutation frequencies as proxy for its relative abundance

(Supplementary Fig. 1). Estimates of the relative abundance of BA.1 produced using the S:L452R dPCR probe were consistently higher compared to those generated using the S:HV69-70 deletion probe or those generated using wastewater NGS (Supplementary Fig. 1). This apparent calibration bias is reflected by the estimates of the midpoint of the curve, which were consistently lower (Supplementary Figs. Y, Z). However, this bias has no discernible effect on the estimates of the growth advantage of the variant (Fig. 3C, Supplementary Fig. 1).

## Discussion

A key concern in wastewater-based epidemiology is the distortion in the relationship between viral load measurements and true incidence due to shedding load profiles, and the possible impact of differential shedding between variants on estimates of incidence. We show here that this is not an issue for the estimation of the selection advantage of a variant. The introduction and spread of a new variant that is shed more or less than the currently circulating variant can introduce bias in the real-time estimates of viral incidence and prevalence, as well as in the estimates of variant relative abundances. However, wastewater-based epidemiology provides other metrics of central importance to public health strategies: the growth rates and the selective advantage of the virus and its variants. Here we show that these quantities are either minimally or not at all sensitive to biases affecting normalization.

In the case of genomic epidemiology, a parameter that is crucial to estimate when a new variant emerges is its selective advantage relative to the currently circulating strain(s)[13]. Here, we have shown that under common assumptions of selection models, estimates of this parameter based on wastewater-derived data are unbiased as well as robust to arbitrary changes in shedding between variants. When relaxing those assumptions, estimates of selection are biased, regardless of whether they are computed from true relative incidence or from observed relative loads. Our results show that some types of changes in shedding profiles, mainly changes in mean shedding time, can amplify this bias in the wastewater-based estimates. Nevertheless, even under relaxed selection model assumptions, we find that differences in total shedding alone, which are the typical changes observed between variants, do not compromise the robustness of selection estimates.

In addition, we studied the effect of combining a change in mean generation time with a change in the shedding profile of a new variant. This reflects scenarios where differences in the infectiousness profiles of variants would affect both shedding and transmission dynamics, such as if altered within-host kinetics lead to earlier or more concentrated infectiousness and shedding. We find that failing to account for differences in mean generation time introduces bias in estimates of selection, whether computed from true relative incidence or from observed relative loads. However, in this setting, the arbitrary change in shedding profile does not amplify the bias. We further used simulations to examine the impact of changes in the shape of the generation time distribution and found no compounding effect from differences in total shedding. However, we did not explore cases where both the shape of the shedding profile and the shape of the generation time distribution vary simultaneously. In such cases, we expect that the effects could compound.

A parameter describing the current disease dynamics, which is critical to assess the epidemiological situation and inform policy, is the effective reproduction number $R_e$. We have shown that the bias of $R_e$ estimation stemming from differences in shedding profiles is transient. That is, the estimator will recalibrate itself after a short period of time. Additionally, we have provided approximate bounds for the maximum bias during this transition period.

Clinical data is often considered the gold standard to obtain epidemiological insights into the dynamics of a pathogen and its variants. However, clinical estimates are also not indicative of the true incidence of the pathogen or its variants because they are confounded by a delay distribution between infection and testing[22]. Additionally, estimates based on clinical data are subject to biases, such as non-random testing and sequencing. In this study, clinical sequences from randomly selected positive tests of the general population were compared to wastewater-derived data. Randomly selected samples should suffer fewer biases related to age stratification and variation in triaging that can be encountered when aggregating non-random samples, such as is typically obtained from hospitals. Nevertheless, the samples remain subject to sampling biases, as the testing was not random[23]. Moreover, estimates based on clinical data require orders of magnitude more samples than needed for wastewater to achieve the same precision, because wastewater samples provided a pooled population estimate. Our findings demonstrate that wastewater-based epidemiology remains a valid (and advantageous) tool to estimate competitive advantages between multiple variants of a pathogen and to assess the current disease dynamics, even in the presence of differential shedding.

Our findings also support that a large number of possible biases stemming from the protocol used for quantifying pathogens in wastewater (e.g., underestimating a variant prevalence due to the dPCR probe used), will also not impact the estimation of the selection advantage. This robustness makes integrating various data sources into a large scale program easier and more robust to differences in methodology among participating labs. Moreover, the results we have presented here are sufficiently general to apply to modeling the selective advantage of strains or variants of other pathogens beyond SARS-CoV-2, highlighting the relevance of the findings for the expansion of wastewater genomic surveillance in the future.

Our study also highlights the potential of tracking SARS-CoV-2 variants using dPCR assays. Tracking the progression of a variant using dPCR relies on the availability of an assay targeting a mutation separating it from the currently circulating variants. Using an already established assay can reduce the lead time needed to track the progression of a new variant. We tracked the spread of BA.1 using not only the rise of a BA.1 signature mutation (S:HV69-70), but also using the decline of a Delta signature mutation (L452R). Despite yielding different estimates of the relative abundance of Omicron BA.1, the same estimates of the selection advantage were obtained using the different dPCR assays. This result further illustrates the robustness of estimating the selection advantage of a variant to quantification or calibration biases.

A limitation of this study is that we have focused here on the bias of our estimators of interest, but the (squared) bias is just one component of the expected (squared) error, the other being the variance. It might very well be that differences in shedding profiles, such as lower shedding or shedding load distributions with higher spread, can add variance to our estimates of interest and thus increase the error. Outside of the scope of this study, but of potential interest, is the extent to which changes in shedding can delay early estimation of parameters of interest due to a loss of precision.

To conclude, we suggest generally that if there is a potential introduction of a new source of error in the quantification of a pathogen or its variants, it is essential to understand the impact of this error on the estimation of derived public health-relevant indicators. As we showed here for differences in shedding, a source of bias may have no bearing on the outcome, such as estimates of growth rates. Alternatively, it might also be that even though error propagates to impact the relevant indicators, the impact is of a small magnitude relative to the already present statistical error, rendering the effect negligible. If feasible and even if based on strong assumptions, closed-form expressions and approximations can clarify the nature of the bias. Additionally, as shown here, simulations can help test the assumptions of closed-form expressions or replace them entirely when they are not available.

## Methods

### Bias and robustness of the selection advantage estimates

We consider the incidence of variants $X$ and $Y$, which vary according to the model (Supplementary A)

$$X'(t) = \beta_x X(t) - \gamma X(t) \text{ and } Y'(t) = \beta_y Y(t) - \gamma Y(t) \quad (3)$$

In this model, $\beta$ represents the transmission rate, i.e., the rate at which infected individuals infect non-infected individuals, and $\gamma$ represents the rate at which infected individuals stop being infectious, yielding a mean generation interval time of $\gamma^{-1}$. We model a constant selection advantage $s$ of variant $X$ over variant $Y$ as $s = \beta_x/\beta_y - 1$, such that $R_x/R_y = 1 + s$, where $R_x$ and $R_y$ are the instantaneous reproduction numbers of $X$ and $Y$.

We consider a general procedure for estimating $s_x$ from relative incidence time series data $f(t) = \frac{X(t)}{X(t)+Y(t)}$. This procedure starts by estimating the time derivative of the logit-transformed relative abundances $\phi(t)$, which give the difference in transmission rates

$$\phi'(t) = \log\left(\frac{f(t)}{1-f(t)}\right)' = \beta_x - \beta_y \quad (4)$$

Commonly, $\phi'$ is estimated by modeling $f(t) \sim t$ using some variant of logistic regression[11–13]. The estimate of $\phi'$ is translated into an estimate of the selection advantage $\hat{s}$ following

$$\hat{s} = \phi'\gamma^{-1} \quad (5)$$

This approach makes some key assumptions. First and foremost, it assumes that the transmission rate of each variant stays constant through time. When they vary (for example due to an intervention or due to a depletion of available susceptible individuals), the estimated selection advantage will appear to vary even though it is constant. Second and third, they assume that the generation times are equal for both variants and that the instantaneous reproduction number for variant $Y$ is close to one. Violations of both of these assumptions can lead to a bias in the estimation of the selection advantage of $X$ over $Y$. Furthermore, the model assumes a large, well-mixed isolated population. Population dynamic effects such as high rates of introduction of a new variant from other locations can confound the analysis of its selection advantage. Lastly, this model is built on the assumption that the selection coefficient $s$ is constant. This model is here formulated for simplicity for two variants, but it generalizes to any number of variants, for example by modelling the relative abundance $X_i/(X_i + X_j)$ for all pairs of variants $X_i, X_j$.

In practice, we do not have access to the true relative incidence $f(t)$ but need to use a proxy, generally based on clinical testing and typing. When using wastewater-derived data, the analysis is complicated by the shedding load profiles $g$ of the virus, a function which represents the average shedding of viral particles through time after infection. When $g$ is normalized by the total amount of particles shed, we speak of the shedding load distribution. In general, the shedding load profiles will scale, shift, and spread the observed viral loads in wastewater relative to the true incidence (Supplementary B, Supplementary Fig. 4). We say that the estimates of the selection advantage $\hat{s}^w$ based on relative loads of variants in wastewater data $f^w(t)$ are unbiased if they have no expected difference from the estimate derived from the true relative incidences. We say that they are robust if the two variants having different shedding load profiles $g_x$ and $g_y$ does not introduce any bias. To investigate the bias and robustness of wastewater-based estimates of relative abundance, we investigate under which conditions the time derivative of the logit transformed

relative loads

$$\phi^w(t)' = \frac{(X*g_x)'(t)}{(X*g_x)(t)} - \frac{(Y*g_y)'(t)}{(Y*g_y)(t)} \quad (6)$$

is altered relative to $\phi(t)'$ (Supplementary C). Here, $X*g_x$ and $Y*g_y$ represent the convolution product of the incidences of variants $X$ and $Y$ with their respective shedding load profiles. With this approach we show that under the common assumption of constant transmission rates, the wastewater-derived estimates are both unbiased and robust to any arbitrary changes of the shedding load profile (Supplementary C i). Furthermore, we also show that if we further assume that $g_x$ and $g_y$ have the same shape, so that only the total amount shed is different, then the estimates of selective advantage are always robust even when transmission rates are not constant (Supplementary C ii). The intuition behind these derivations is that, if the observed loads correspond to scalings of the true incidences, then the logit transformed relative loads $\phi^w(t)$ will be equal to the logit transformed relative incidence $\phi(t)$ up to an additive constant, and therefore have the same time derivative. For showing that this applies to arbitrary shedding profiles in the case where transmission rates are constant, we can use a key property of exponential growth where convolution with a distribution is equivalent to a scaling.

We also use this framework to assess the impact of incorrectly assuming identical mean generation times and shedding profiles for both variants. In this case, estimates derived from wastewater data are affected in the same way as those based on true relative incidence (Supplementary C viii). Similarly, we use the same framework to show that if we wrongly assume a constant generation time, then the estimates based on wastewater data are affected in the same way as estimates based on true relative incidence, although the bias might be delayed in time (Supplementary C v). We also show that in that case the wastewater-based estimates remain robust to changes in total shedding.

We further develop approximations of $\phi^w(t)'$ to investigate the eventual bias in the apparent selective advantage when the assumption of constant transmission rates does not hold or when the assumption of constant selection does not hold (Supplementary C iii, iv). We use it to investigate its eventual loss of robustness when, in addition, $g_x$ is arbitrarily different in shape from $g_y$ (Supplementary C vi). These approximations show that with non-constant transmission rates or with a non-constant selection factor $s(t)$, the wastewater-based estimates $\hat{s}^w(t)$ are delayed with respect to true relative incidence-based estimator $\hat{s}(t)$ by an amount equal to the mean $\mu$ of the shedding load distribution $g$ (although it is still robust to changes in total shedding from a variant).

$$\hat{s}^w(t) \approx \hat{s}(t - \mu) \quad (7)$$

When we additionally further relax the assumption that $g_x$ and $g_y$ have the same shape, then the approximations show that robustness can be lost, and that the main contributing factors are the difference in mean of the shedding load distributions of $X$ and $Y$, and the rate at which the transmission rates are varying (Supplementary Equations 38, 39).

### Bias in $R_e$ estimates due to differences in shedding

We derived closed-form expressions to quantify the potential bias in wastewater-based estimates of $R_e$ during the introduction of a variant with a different shedding load profile (Supplementary D). For the case where the shedding load profiles of the variants differ in total shedding

by a factor $c$, we obtained (Supplementary D ii) the additive bias

$$R^w(t) = R(t) + \gamma^{-1} \frac{(c-1)f(t)'}{1+(c-1)f(t)} \leq \frac{(c-1)s}{4} \tag{8}$$

In addition, when the shedding load distributions differ (Supplementary D iii), we have the bias

$$R^w(t) \approx R(t) + \gamma^{-1} \frac{(\beta_x - \gamma)\Delta_\mu f(t)'}{1+(\beta_x-\gamma)\Delta_\mu f(t)} \leq \frac{s(\beta_x-\gamma)\Delta_\mu}{4} \tag{9}$$

Where $\Delta_\mu = \mu_y - \mu_x$. This also shows that the bias is greatest when the relative abundance $f(t)$ changes rapidly over time, but goes to zero if $f(t)$ does not vary over time (such as at the end of a sweep).

## Simulations of variant emergence

We performed deterministic simulations of variant emergence for four different scenarios, two with fixed transmission rates and two with varying transmission rates. First, two variants are introduced at the same time and compete, with one having a selective advantage. Second, one variant is going extinct and is swept by the second variant. Third, two variants are introduced at the same time, but their transmission rates decrease according to the mass action dynamics of a Susceptible-Infected-Recovered (SIR) model. Fourth, two variants are introduced at the same time, and the transmission rates are reduced to almost zero by an intervention in the middle of the time series. In all cases, we assumed a fixed multiplicative selective advantage. We performed the simulations in R v4.1.3 using the package *deSolve v1.35*[24].

In the derivations of closed-form expressions for the biases in selection advantage $s$ and the effective reproduction number $R_e$ due to reduced or increased shedding, we relied on an expression for $R_e$ which assumes an exponentially-distributed generation time interval. We relaxed this assumption in stochastic simulations to assess the robustness of our results, using the simulation framework used in Huisman et al. [2,23], based on the framework described by Cori et al. [23,25] (Supplementary D). In this case the generation time interval is assumed to be gamma distributed, with a mean of 4.8 days and a standard deviation of 2.3 days. The results were analyzed using *EpiEstim v2.2*[26] using a sliding window approach.

## Wastewater sampling and processing

Wastewater-based surveillance for SARS-CoV-2 was conducted in six sewersheds (Altenrhein, Chur, Geneva, Laupen, Lugano, Zurich) across Switzerland from 24 November 2021 through 10 January 2022 (Fig. 3A, Supplementary Fig. 1). Volume-proportional 24 h composite samples were collected daily from raw wastewater at the influent of each of the six sites and stored at 4 °C for up to 5 days before being transported on ice for processing at a central laboratory (Eawag, Dübendorf, Switzerland). Processing included total nucleic acid extraction from 40 ml samples (Wizard Enviro Total Nucleic Acid Extraction Kit, CN A2991, Promega Corporation, USA) with an elution volume of 80 ul and subsequent inhibitor removal using OneStep PCR Inhibitor Removal columns (CN D6030, Zymo Research, USA). From these samples, a subset were analyzed for variants of concern using drop-off RT-dPCR assays targeting signature mutations for Delta (S:L452R, $n = 74$ samples) and Omicron BA.1 (S:HV69-70, $n = 79$), following Caduff et al. [4]. Almost all ($n = 280$) samples were also analyzed using NGS to identify Delta versus Omicron BA.1, following Jahn et al. [5]. RNA extracts were stored at −80 °C for up to 1 week prior to sequencing and up to 3 months prior to RT-dPCR analysis.

## Drop-off RT-dPCR assays for detection of signature mutations

Drop-off RT-dPCR assays targeted the S:HV69-70 deletion indicative of Omicron BA.1, as previously described[4], and the S:L452R mutation indicative of Delta (lineage B.1.617.2).The assay targeting S:L452R includes two hydrolysis probes binding to a single amplicon: a universal probe targeting a conserved region on the amplicon and a variant-specific probe that binds to the S:L452R mutation (Supplementary Table 3, Supplementary Fig. 3). In the digital PCR, generated droplets with dual fluorescence indicates the presence of an amplicon from the mutation (i.e., Delta), whereas single fluorescence indicative of only the universal probe indicates an amplicon without the S:L452R mutation (i.e., Omicron). For the S:HV69-70 assay, the variant-specific probe only binds when S:HV69-70 is present[27], and therefore dual fluorescence in a given droplet indicates presence of an amplicon without the mutation (i.e. Delta), whereas single fluorescence of only the universal probe indicates the presence of the mutation (i.e., Omicron).

## Clinical VOC data

For each canton surrounding a WWTP, we downloaded counts of infected individuals binned by variant of sequenced PCR-positive clinical samples through the LAPIS API of CoV-Spectrum[28]. The data was restricted to sequences originating from the Viollier lab (8525 sequences, Supplementary Fig. 1), which during that period sent a random subset of their PCR-positive samples out for sequencing. This ensured that we compared to the most random clinical samples available for the populations served by the WWTP we analyzed. For the Geneva sewershed, we also compared estimates to data on the qPCR S-gene target failure (SGTF) data from the Geneva University Hospital (HUG) available at https://www.hug.ch/laboratoire-virologie/surveillance-variants-sars-cov-2-geneve-national (5794 tests, Supplementary Fig. 1). The SGTF qPCR is based on the detection of S:HV69-70; failed amplification of a clinical sample previously positive for the N1 gene target is used as a proxy to indicate that the clinical sample is Omicron BA.1.

## Data analysis

Computational analysis of wastewater sequencing samples was performed using V-Pipe 3.0[29]. We analyzed the data using the R v4.1.3 statistical programming language and the R package WWdPCR v0.1.0[14]. The package was used to obtain maximum likelihood estimates (MLE) and confidence intervals of the logistic growth rate of the Omicron BA.1 variant in each region. Confidence intervals for the logistic growth parameter were computed assuming a quasibinomial (for the clinical and wastewater sequencing data) or quasimultinomial (for the dPCR data) model of the counts to account for overdispersion, but without allowing underdispersion (i.e., overdispersion factors <1 were not considered). Confidence bands for the fitted values were computed on the logit scale, with standard errors projected using the Delta method, and then back-transformed to the linear scale. The normality assumption is more likely to hold in this reparametrization, which optimizes the coverage of the intervals as well as ensures that they are constrained to the [0,1] range[30].

The logistic growth model was fitted separately using the wastewater S:L452R dPCR data, the wastewater S:HV69-70 dPCR data, the wastewater sequencing data, and the clinical sequencing data. For Geneva, we fit the model also on the SGTF data. For the S:L452R dPCR, the dual fluorescence droplets are indicative of Delta, so we assumed that the single fluorescence droplets indicated Omicron BA.1. The growth advantage of BA.1 was calculated using S:L452R data assuming a negative logistic decay rate of the mutated fraction. For wastewater sequencing, we proceeded as previously described[5], and we estimated the relative abundance of BA.1 using the observed fractions of reads bearing mutations characteristic of BA.1, while excluding mutations also present in B.1.617.2*. Amplicons suffering potential differential dropout rates or altered amplification due to mutations in the primer regions were discarded.

## Inclusion & Ethics statement

All collaborators who contributed to this study and fulfilled the authorship criteria required by Nature Portfolio journals have been included as authors.

## Reporting summary

Further information on research design is available in the Nature Portfolio Reporting Summary linked to this article.

## Data availability

All data necessary to reproduce the results are available at https://doi.org/10.5281/zenodo.10040850. Sequencing data from the Swiss wastewater surveillance program are available on the European Nucleotide Archive (ENA) under project accession number PRJEB44932. Counts of clinical sequences were accessed through the LAPIS API of CoV-Spectrum (https://lapis.cov-spectrum.org/), and the qPCR S-gene target failure (SGTF) data from the Geneva University Hospital (HUG) is available at https://www.hug.ch/laboratoire-virologie/surveillance-variants-sars-cov-2-geneve-national. Both sources of clinical data are public, do not require approval for access and do not contain patient-level information.

## Code availability

All code necessary to reproduce the results is available at https://doi.org/10.5281/zenodo.10040850.

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

## Acknowledgements

This study was supported by the Swiss National Science Foundation (grant no. CRSII5_205933, N.B., T.R.J., T.S., C.O.) and the Swiss Federal Office of Public Health. We thank the members of the Wastewater-based Infectious disease SurveillancE (WISE) consortium. We thank the operators of the wastewater treatment plants for providing samples.

## Author contributions

Conceptualization: D.D., J.S.H., N.B., T.R.J.; Methodology: D.D., J.S.H., J.C.R., L.C., P.G., A.J.D., C.G., T.R.J.; Software: D.D., J.S.H.; Validation: D.D., J.C.R., L.C.; Formal analysis: D.D.; Investigation: D.D., J.S.H., J.C.R., L.C., P.G., A.J.D., C.G.; Resources: T.S., T.K., C.O., N.B., T.R.J.; Data curation: D.D.; Writing – original draft: D.D.; Writing – review & editing: all authors; Visualization: D.D.; Funding acquisition: T.S., T.K., C.O., N.B., T.R.J.; Supervision: N.B., T.R.J., J.S.H.

## Competing interests

The authors declare no competing interests.
