## [Transparent Peer Review file · Nature Communications]

Estimated transmission dynamics of SARS-CoV-2 variants from wastewater are unbiased and robust to differential shedding

Corresponding Author: Professor Niko Beerenwinkel

Version 0:

Reviewer comments:

Reviewer #1

(Remarks to the Author)

Review of the manuscript 'Estimated transmission dynamics of SARS-CoV-2 variants from wastewater are robust to differential shedding' by David Dreifuss and colleagues, submitted to Nature Communications.

Summary

In this manuscript, the authors aim to gauge the impact of differential shedding profiles of SARS-CoV-2 variants on estimates of growth differences between variants and effective reproduction numbers using analyses of wastewater data. The manuscript consists of two parts. In the first, the authors present and analyse models to yield theoretical expectations using a mesh of theoretical analysis and computer simulations (Figure 2, sections on "Bias in growth advantage ..." and "Stochastic simulations of variant emergence" in the Methods). In the second, the authors apply the methods to data from six wastewater treatment plant in Switzerland in the short period covering the transition from the Delta to the first Omicron variant (Figure 3). The authors claim that the theoretical analyses show that growth rate estimates are invariant to differences in shedding, that differences in shedding have a transient and small effect on estimates of the effective reproduction number, and that the distribution of the generation time does not affect estimates of growth advantages of newly arising variants. The application to the Delta-to-Omicron transition show that, in general, various estimates of growth rates (including those based on testing data) are of similar magnitude.

Evaluation

The topic of analyses is of substantial interest and of value for the interpretation of wastewater-based SARS-CoV-2 epi analyses. However, I have several important reservations, mentioned below.

1) First and foremost, I am convinced that there is an error in the equation on the top of page 9 (last equality). In fact, the left-hand reads $\log((X * g)(t) / (Y * g)(t))$ and the right-hand reads (after inserting $f(t) = X(t) / (X(t) + Y(t))$) $\log(X(t) / Y(t))$. Those two can obviously only be true for very specific functions X, Y and/or g. Making use of the fact that Y scales to X by a constant c does not solve the problem. It seems as if the authors have simply crossed out g in the convolution of the left-hand of the equation. And this, in turn, is probably due to sloppy use of notation for convolutions suggesting that the convolution is a simple product of time-dependent functions (which it is not). In fact, I strongly suggest to adhere to more standard notation, such as $(X * f)(t)$. The mistake is not without consequences, as it invalidates the first Result ("Growth advantage estimates are invariant to differences in shedding").

2) Similarly, I found the derivation of the equation for the derivative of R(t) on page 9 unclear, especially the step from the second to the third line. This may be ok but more explanation is needed on the assumption involved in the derivation.

3) I found it insufficiently clear to which extent the general theoretical results depend on model assumptions (fixed shedding shape, no true transmission model and interval contraction). In addition, I found the actual results on the Swiss data lacking in discussion and sensitivity analyses (does the method yield similar results with hospital data? Can trends in Fgi3b be explained by testing practices? What would the impact be of variable testing practices? What about the potential

consequences of varying severity of infection leading to varying reporting rates?). Further, I found the discussion somewhat lacking in depth. For instance, would these analyses also work for other strain transitions? What are the intuitive explanations for the theoretical results (in so far as they remain after revision). What if more than two strains are circulating? Can this be included in the framework? Are these methods applicable to other pathogens that can be quantified in sewage?

Some specific comments

-p1 third paragraph, please give intuition of the prevalence versus incidence argument (ref 13).

-p1 third paragraph. I understand that you would prefer to focus on scenarios in which the shape of the excretion curve remains constant. And I agree that the Puhach paper (ref 14) suggests that this is the case for loads in the respiratory tract. But this is not necessarily the case for fecal excretion, and I would have expected an analysis in the simulation study that lets go of this strong assumption.

-Fig3b. Could the decrease be due to a testing bias? Please elaborate and discuss possible implications. Why did you not use hospital data, as these are probably less prone to such bias?

-How were the EpiEstim analyses performed? More specific information is needed.

-Confidence bands with delta method formally only apply if the data are normally distributed. Please explain/elaborate.

-p9 simulation study. As far as I can see these simulations do not make use of a transmission model, but rather take generation interval distributions for two strains and reproduction numbers to simulate case. If this is the case, the analyses in essence assume an infinitely large pool of susceptible individuals and no competition for susceptibles, ie it describes a branching process framework? Is this so? In any case, I think this section needs more/better explanation.

-Fig S3 also needs more/better explanation and guidance for the reader.

Recommendation

Based on the above issues I cannot recommend the current manuscript for publication in Nature Communications. Given the importance of the topic I could envisage a thoroughly reworked manuscript, that deals with the above error, makes it much clearer to what extent formal derivations depend on model assumptions, focuses in more detail on the actual data (possibly including other strain transitions as well), and provides a more in-depth discussion of the results could be resubmitted.

(Remarks on code availability)

I have had a brief look at the code, which looked reasonably well structured.

Reviewer #2

(Remarks to the Author)

The authors presented a mathematical modelling study to evaluate whether differential shedding profiles would affect the estimation of transmission advantage of different SARS-CoV-2 variants of concern using wastewater surveillance data. The methods were applied to Swiss surveillance data as a case study. While I do think the research question is important and highly relevant, given the rapid development of wastewater-based epidemiology during the COVID-19 pandemic, some of the assumptions used in the simulation and validation seem to be oversimplified. Thus I have reservation about the statement claiming that "the bias does not affect estimation of the growth advantage of the variant and has only a limited and transient impact". Please see below for my specific comments.

1. The assumption on the constant parameter c seems to be oversimplified. Differential shedding profiles could be shown in the distribution of viral shedding in various format, with the area under the curve of the shedding distribution reduced by a proportion of c . However, in the methods, the authors only evaluated one possible scenario with the shedding distribution of one variant as $g(t)$ and the other as $cg(t)$. More scenarios should be taken into account instead of simply assuming the same increase or reduction in shedding over time in the shedding distribution.

2. Following the comment above, the generation time distribution would be potentially affected by the differential shedding profiles. And the shedding profile is an indicator of the infectiousness. Therefore, it is prudent to not only consider the difference in the effective reproductive number Re (e.g. $Re = 0.6$ and 2.2 in the simulation) but also the difference in the generation time distribution (e.g. mean of 4.8 days and variance of 5 days²). What about the scenario with the Re of 0.6 vs 2.2 and mean generation time of 4.8 vs 2.4 days?

3. Figure 2: It is hard to tell whether the bias in the estimation of growth advantage is transient from the results presented in Figure 2. How long in time did the two variants co-circulate (e.g., with both variants exceeding 5% of the daily new infections in the simulation)? Or the timing of the later variant exceeding 5%, 50% and 95% of the daily new infections? The estimation of growth advantage should be affected most when both variants take up substantial proportion among the daily new infections. If the major co-circulation period is only a week, a bias of two days would be considered substantial.

4. Figure 3: Similarly, for the six locations from the Swiss surveillance, how were the bias considered large or not? Was the growth advantage estimated from the clinical surveillance data taken as the "ground truth"? It would be helpful to map the relative abundance in Figure S1 and the estimated bias in growth advantage over time. It is hard to tell whether the bias is

transient from the current Figure 3.

5. Assuming clinical surveillance data are complete and accurate from the simulation, wouldn't the integration of both clinical and wastewater surveillance data improve the estimation of growth/transmission advantage? If so, could the authors try a few simulations and compare the performance of estimation with and without clinical data?

(Remarks on code availability)

Reviewer #3

(Remarks to the Author)

Dr Dreifuss and colleagues presented an interesting article on the estimation of the transmission dynamics of SARS-Cov-2 (that could be held as a case study for similar pathogens) in wastewater. I particularly appreciated the efforts made to investigate mathematically and computationally potential sources of bias that may impact the effectiveness of wastewater surveillance.

However, I would like authors to strengthen their results by including additional analysis. In particular the difference in shedding is assumed to be a constant factor rescaling the incidence. This seems a rather simple assumption (considering that the shedding profile could be different due to lower days of shedding, lower viral load or a combination of both) and should be better highlighted and discussed if not revised. For instance in the stochastic simulations of variant emergence. The design of this analysis is not clear to me as it seems to be applied to the number of cases rather than to wastewater samples. If I understood correctly authors simulated two time series of cases and then to simulate undershedding they scaled down by 50% the time series of the emerging variants. This approach could also be applied (and has been applied) to simulate underreporting (ie the emerging variant being less symptomatic) and similar results have been already discussed in the context of underreporting. However it does not take into account that what is observed in wastewater surveillance is not the number of cases, but a viral load, hence missing an intermediate step. I think it would be more interesting if authors simulated the time series of cases and then from the daily number of cases generated the potentially observed values in wastewater data by exploiting a shedding profile (that could differ between variants). In this way the cumulative effects of a case shedding for half the days or shedding half the dose due to being infected by a different variant would be correctly investigated in the frame of wastewater surveillance (that is blind to the actual number of cases).

Another aspect that needs revision is the assumption of a constant performance of the sewage system in maintaining viral load. This assumption is not adequately discussed in the manuscript as the whole mathematical analysis assumed that the incidence in the population is unambiguously measured in the population (by wastewater samples) while different dilutions (or other factors) could bias this estimate.

(Remarks on code availability)

Version 1:

Reviewer comments:

Reviewer #1

(Remarks to the Author)

(Remarks on code availability)

Reviewer #2

(Remarks to the Author)

Thank you for the opportunity to review the revised manuscript. My main comments are:

1. My previous comment R2.2 was not adequately addressed. While the manuscript examined the impacts of viral shedding profiles and generation time on wastewater-derived epidemiological estimates, it largely treated these as independent factors. The paper would be strengthened by explicitly discussing and potentially modeling the intrinsic biological link between shedding patterns (timing and magnitude) and an individual's infectiousness profile, as this profile dictates generation time and likely correlates with shedding.

2. The analysis tested sensitivity to generation time dispersion but should have further explored how altered shedding profiles, if mechanistically linked to infectiousness, could systematically change the generation time distribution (both mean and shape). It is unclear if the current robustness checks for generation time fully capture these coupled dynamics.

3. The conclusion that estimates of selection advantage are robust to separate differences in total shedding and generation times warrants further discussion. The authors should address the implications if shedding characteristics and infectiousness profiles (and thus generation times) are coupled—for example, if a variant exhibits a different shedding profile due to an altered infectiousness course. The potential impact of such biologically plausible couplings on the robustness of the findings needs more thorough consideration.

(Remarks on code availability)

Reviewer #3

(Remarks to the Author)

I am satisfied with the responses, and have no further comments. Thank you

(Remarks on code availability)

Reviewer #4

(Remarks to the Author)

(Remarks on code availability)

Version 2:

Reviewer comments:

Reviewer #1

(Remarks to the Author)

I am happy with the latest set of revisions.

(Remarks on code availability)

Reviewer #2

(Remarks to the Author)

Thank you for inviting me to review the revised manuscript again. The manuscript improved significantly after explicitly accounting for individual shedding in the wastewater-derived epidemiological estimates. I concur with Reviewer 1's comment that the manuscript would benefit from a more balanced discussion regarding the robustness of the fitness estimates.

With caution, the statement "Another standard assumption in evolutionary models used to estimate the fitness advantages of SARS-CoV-2 variants is that variants have the same generation time" should not be presented as a "standard" assumption. Therefore, the subsequent claim (and throughout the manuscript), "Here, we have shown that under standard assumptions of selection models, estimates of this parameter based on wastewater-derived data are unbiased as well as robust to arbitrary changes in shedding between variants," is less convincing. I suggest that the authors refer to this assumption as the "baseline" instead of claiming it as a "standard assumption."

(Remarks on code availability)

Reviewer #4

(Remarks to the Author)

(Remarks on code availability)

Response to reviewers

Reviewer #1 (Remarks to the Author):

Review of the manuscript 'Estimated transmission dynamics of SARS-CoV-2 variants from wastewater are robust to differential shedding' by David Dreifuss and colleagues, submitted to Nature Communications.

Summary

In this manuscript, the authors aim to gauge the impact of differential shedding profiles of SARS-CoV-2 variants on estimates of growth differences between variants and effective reproduction numbers using analyses of wastewater data. The manuscript consists of two parts. In the first, the authors present and analyse models to yield theoretical expectations using a mesh of theoretical analysis and computer simulations (Figure 2, sections on "Bias in growth advantage ..." and "Stochastic simulations of variant emergence" in the Methods). In the second, the authors apply the methods to data from six wastewater treatment plant in Switzerland in the short period covering the transition from the Delta to the first Omicron variant (Figure 3). The authors claim that the theoretical analyses show that growth rate estimates are invariant to differences in shedding, that differences in shedding have a transient and small effect on estimates of the effective reproduction number, and that the distribution of the generation time does not affect estimates of growth advantages of newly arising variants. The application to the Delta-to-Omicron transition show that, in general, various estimates of growth rates (including those based on testing data) are of similar magnitude.

Evaluation

The topic of analyses is of substantial interest and of value for the interpretation of wastewater-based SARS-CoV-2 epi analyses. However, I have several important reservations, mentioned below.

R1.1. First and foremost, I am convinced that there is an error in the equation on the top of page 9 (last equality). In fact, the left-hand reads $\log((X * g)(t)/(Y * g)(t))$ and the right-hand reads (after inserting $f(t)=X(t)/(X(t)+Y(t))$) $\log(X(t)/Y(t))$. Those two can obviously only be true for very specific functions X , Y and/or g . Making use of the fact that Y scales to X by a constant c does not solve the problem. It seems as if the authors have simply crossed out g in the convolution of the left-hand of the equation. And this, in turn, is probably due to sloppy use of notation for convolutions suggesting that the convolution is a simple product of time-dependent functions (which it is not). In fact, I strongly suggest to adhere to more standard notation, such as $(X * f)(t)$. The mistake is not without consequences, as it invalidates the first Result ("Growth advantage estimates are invariant to differences in shedding").

Thank you very much for your thorough review. You are correct that there was a mistake in the way our results were presented, particularly in the equation on page 9. We apologize for the confusion. We have now corrected this error, and Supplementary equations 19-21 now depict

$$\log ((X * cg)(t)/(Y * g)(t)) = \log ((X * g)(t)/(Y * g)(t)) + \log (c)$$

Notably with the correction, the original claims of the manuscript, i.e. the selection advantage estimates are robust to scaling changes in the shedding load profiles, still stand.

Additionally, we have also adopted the notation you suggested for convolution products, to make the expressions clearer and less ambiguous. These changes have been incorporated throughout the manuscript, improving the presentation of our results.

R1.2. Similarly, I found the derivation of the equation for the derivative of R(t) on page 9 unclear, especially the step from the second to the third line. This may be ok but more explanation is needed on the assumption involved in the derivation.

Thank you for your feedback. We have fully revised the derivation to improve clarity and have included more intermediate steps to better highlight the underlying assumptions. The extended derivation can be found in Supplementary D, Supplementary Equations 45-54.

R1.3.a. I found it insufficiently clear to which extent the general theoretical results depend on model assumptions (fixed shedding shape, no true transmission model and interval contraction).

This was a concern raised by all reviewers, so we have significantly expanded our results section. We have clarified our transmission models and the assumptions underlying them (Equation 3, Supplementary A). Additionally, we have now thoroughly explored the effects of not only varying total shedding, but also the effects of arbitrary changes in shedding shape (Figure 2, Supplementary Table 1, Supplementary C). We also investigated the impact of relaxing key assumptions, including constant infection rates, constant mean generation times and constant selection advantage. These revisions have strengthened our findings considerably, and we appreciate your suggestion in helping us enhance the robustness of our results.

R1.3.b. In addition, I found the actual results on the Swiss data lacking in discussion and sensitivity analyses (does the method yield similar results with hospital data? Can trends in Fgi3b be explained by testing practices? What would the impact be of variable testing practices? What about the potential consequences of varying severity of infection leading to varying reporting rates?).

Thank you for this comment. We clarified in the results and methods sections that the clinical data available was as randomly sampled as possible, specifically including clinical sequences obtained from randomly sampled positive tests from a large testing company, along with data from hospital surveillance. We now also address in the discussion how varying reporting rates in clinical data are accounted for by our results.

Additionally, it is a valid and very interesting point that a reduction in the ratio of wastewater viral loads per new confirmed case could be influenced by changes in testing practices. We added two figures (Supplementary Figure 6) which show that the test positivity rate increased during the period considered. An increase in test positivity rate is generally assumed to be indicative of an increase in underreporting of the incidence. These additional data show that testing practice likely did not

explain the effect we observed. On the contrary, the ratio of viral loads per case (i.e., the total fecal shedding per case) could have declined even more than what was apparent using the reported positive cases for normalization.

R1.3.c. Further, I found the discussion somewhat lacking in depth. For instance, would these analyses also work for other strain transitions? What are the intuitive explanations for the theoretical results (in so far as they remain after revision). What if more than two strains are circulating? Can this be included in the framework? Are these methods applicable to other pathogens that can be quantified in sewage?

We have reworked the discussion to address this comment. We added to the discussion that these results are general enough to be applicable to other pathogens that can be quantified in sewage, as in our derivations and simulations we made no assumption specific to the dynamics of SARS-CoV-2. Additionally, we clarified that they are applicable to scenarios involving more than two circulating strains.

Some specific comments

R1.4. p1 third paragraph, please give intuition of the prevalence versus incidence argument (ref 13).

Thank you for your suggestion. We have added additional intuition to clarify the prevalence versus incidence argument. The main idea behind the argument we source from Hoffman et al. (ref 15) is that shedding decay throughout the course of infection is believed to be fast, such that most of the shedding occurs close to the time of infection.

R1.5. p1 third paragraph. I understand that you would prefer to focus on scenarios in which the shape of the excretion curve remains constant. And I agree that the Puhach paper (ref 14) suggests that this is the case for loads in the respiratory tract. But this is not necessarily the case for fecal excretion, and I would have expected an analysis in the simulation study that lets go of this strong assumption.

Thank you for this valuable suggestion. As mentioned in R1.3.a., we have now studied the relaxation of this assumption and explored scenarios where shedding varies arbitrarily, rather than assuming a fixed shape of the excretion curve (Figure 2, Supplementary Tables 1,2). In this general setting, we have also found closed-form results and approximations for the bias, depending on additional assumptions of the selection model (Supplementary Equation 18, 34, 35). These additional analyses have strengthened our results, and we appreciate your input in guiding this important addition.

R1.6. Could the decrease be due to a testing bias? Please elaborate and discuss possible implications. Why did you not use hospital data, as these are probably less prone to such bias?

As discussed in R1.3.b., We added plots (Supplementary Figure 6) that show that the test positivity rate increased during the period considered, such that the magnitude of the decrease in the ratio of viral loads per positive case we estimated is most probably on the conservative side.

Additionally, we have included hospital data from one of the regions connected to a wastewater treatment plant analyzed in our study (Supplementary Figure 1). However, we primarily focused on comparing genomic data from sequences generated from a random sample of positive tests from a large testing company in Switzerland. We expect these data to be less biased by the strong age stratification of people seeking hospital treatment for COVID, as well as by variations in triage procedures during the epidemic. We agree that this is an important point to consider, and we have added a discussion on this issue in the manuscript.

R1.7. How were the EpiEstim analyses performed? More specific information is needed.

Thank you for your comment. We have added more information about the EpiEstim analyses in the Methods section. Additionally, the code used for these analyses is provided along with the manuscript.

R1.8. Confidence bands with delta method formally only apply if the data are normally distributed. Please explain/elaborate.

We have clarified in the methods that the confidence bands were computed for the predictor on the logit scale, and then back-transformed to the linear scale. As detailed in Held and Sabanes-Bové (ref 30), this approach generally provides better frequentist coverage in confidence intervals for proportion data close to the boundaries of $[0,1]$ on the linear scale, and ensures they remain within the $[0,1]$ range. Additionally, this approach makes no normality assumption on the distribution of the data, although it assumes asymptotic normality of the estimator.

R1.9. -p9 simulation study. As far as I can see these simulations do not make use of a transmission model, but rather take generation interval distributions for two strains and reproduction numbers to simulate case. If this is the case, the analyses in essence assume an infinitely large pool of susceptible individuals and no competition for susceptibles, ie it describes a branching process framework? Is this so? In any case, I think this section needs more/better explanation.

Thank you for the comment. In that particular simulation experiment, we explored the robustness of our results to varying the generation interval time. We indeed assume no limitation on the rates by the available number of susceptibles. More generally, we have made that assumption clearer in the manuscript, along with other assumptions made in the models we consider (Equation 3, Supplementary A). Additionally, we have explored in this updated version of the manuscript letting go of this assumption and determine what effect it has on the bias and robustness of selection advantage estimates from wastewater data (Figure 2, Supplementary Table 1). These additional results show that, assuming a shedding with similar shape between variants, letting go of the assumption of a very large pool of susceptibles does not affect the robustness of the estimates of selection advantage to changes in total shedding. We also show that, as discussed for example in R1.5., simultaneously letting go of these assumptions on shedding shape and epidemiological dynamics can affect the robustness of the estimator to changes in shedding (Figure 2, Supplementary Table 1), and we provide closed-form approximations (Supplementary Equations 34, 35).

R1.10. -Fig S3 also needs more/better explanation and guidance for the reader.

Thank you for your feedback. We have expanded the explanation and provided more details to enhance the clarity of the figure.

Recommendation

Based on the above issues I cannot recommend the current manuscript for publication in Nature Communications. Given the importance of the topic I could envisage a thoroughly reworked manuscript, that deals with the above error, makes it much clearer to what extent formal derivations depend on model assumptions, focuses in more detail on the actual data (possibly including other strain transitions as well), and provides a more in-depth discussion of the results could be resubmitted.

We deeply appreciate the thorough review and thoughtful feedback on our manuscript. We are grateful for your recognition of the importance of this topic and hope to have addressed all the concerns raised.

Reviewer #1 (Remarks on code availability):

I have had a brief look at the code, which looked reasonably well structured.

Reviewer #2 (Remarks to the Author):

The authors presented a mathematical modelling study to evaluate whether differential shedding profiles would affect the estimation of transmission advantage of different SARS-CoV-2 variants of concern using wastewater surveillance data. The methods were applied to Swiss surveillance data as a case study. While I do think the research question is important and highly relevant, given the rapid development of wastewater-based epidemiology during the COVID-19 pandemic, some of the assumptions used in the simulation and validation seem to be oversimplified. Thus I have reservation about the statement claiming that “the bias does not affect estimation of the growth advantage of the variant and has only a limited and transient impact”. Please see below for my specific comments.

R2.1. The assumption on the constant parameter c seems to be oversimplified. Differential shedding profiles could be shown in the distribution of viral shedding in various format, with the area under the curve of the shedding distribution reduced by a proportion of c . However, in the methods, the authors only evaluated one possible scenario with the shedding distribution of one variant as $g(t)$ and the other as $cg(t)$. More scenarios should be taken into account instead of simply assuming the same increase or reduction in shedding over time in the shedding distribution.

Thank you for your comment. We have now clarified that we focused on differences in total shedding because we believe these are the most relevant for two reasons: they are the variations we mostly expect from experimental data (see for example ref. 16), and they have the greatest impact on the relationship between incidence and loads (Equation 1).

However, we agree with your point (which was also raised by the other reviewers, see for example our answers to R1.3.a. and R1.5.) that exploring differences in the shape of the shedding load distribution is also important. Following your recommendation, we have now thoroughly explored the effects of not only varying total shedding, but also the effects of arbitrary changes in shedding shape (Figure 2, Supplementary Tables 1,2, Supplementary C, D). We appreciate your suggestion, as it has helped to strengthen our findings.

R2.2. Following the comment above, the generation time distribution would be potentially affected by the differential shedding profiles. And the shedding profile is an indicator of the infectiousness. Therefore, it is prudent to not only consider the difference in the effective reproductive number R_e (e.g. $R_e = 0.6$ and 2.2 in the simulation) but also the difference in the generation time distribution (e.g. mean of 4.8 days and variance of 5 days²). What about the scenario with the R_e of 0.6 vs 2.2 and mean generation time of 4.8 vs 2.4 days?

Thank you very much for this comment. Following your suggestion, we have studied changes in generation time and added them to the results, which show that the wastewater-based estimates of selection advantage are still unbiased and robust w.r.t shedding when generation times differ between variants (Supplementary Equation 44). Additionally, we have also studied the effects of generation times which vary over time, and over both variants and time simultaneously. These additional results show that the wastewater-based estimates of selection advantage are unbiased and robust w.r.t shedding when generation time is not constant through time (Supplementary Equations 31,32). When they are both different between variants and varying through time as well, we show that they are robust only to differences in total shedding (Supplementary Equations 41-43). We deeply appreciate your comment, as it has helped generalize our findings.

R2.3. Figure 2: It is hard to tell whether the bias in the estimation of growth advantage is transient from the results presented in Figure 2. How long in time did the two variants co-circulate (e.g., with both variants exceeding 5% of the daily new infections in the simulation)? Or the timing of the later variant exceeding 5%, 50% and 95% of the daily new infections? The estimation of growth advantage should be affected most when both variants take up substantial proportion among the daily new infections. If the major co-circulation period is only a week, a bias of two days would be considered substantial.

Thank you for your comment. We have added contour lines to the plot (which is now in Supplementary Figure 5) indicating when the variant reaches 5%, 25%, 75%, and 95% of infections. This addition helps to visualize how long the bias in the estimation of the effective reproduction number persists before vanishing.

R2.4. Figure 3: Similarly, for the six locations from the Swiss surveillance, how were the bias considered large or not? Was the growth advantage estimated from the clinical surveillance data taken as the "ground truth"? It would be helpful to map the relative abundance in Figure S1 and the estimated bias in growth advantage over time. It is hard to tell whether the bias is transient from the current Figure 3.

Thank you for your comment. We have clarified in the manuscript that for the selection advantage, which is what we estimate from these data here, we expect the bias to be zero (and not transient). In this analysis, we compare the estimates from wastewater to those obtained from clinical data from the populations served by the treatment plants in the relevant cantons. Indeed, the clinical data estimates are not absolute ground truth, but they are considered the "gold standard" for obtaining such estimates. We have also clarified the criteria that we have used to select the clinical data we have used for this analysis.

R2.5. Assuming clinical surveillance data are complete and accurate from the simulation, wouldn't the integration of both clinical and wastewater surveillance data improve the estimation of growth/transmission advantage? If so, could the authors try a few simulations and compare the performance of estimation with and without clinical data?

Thank you for your comment. While integrating both clinical and wastewater surveillance data into the same model could have a chance to improve the estimation of growth or transmission advantage, this approach would require entirely new method development, which is beyond the scope of this paper. Additionally, it is important to note that if clinical data were truly complete and accurate, the need for wastewater data would diminish, as we would already have all the necessary information. However, in reality, clinical data is often incomplete and costly to collect comprehensively, which is why wastewater surveillance provides a valuable and cost-effective alternative.

Reviewer #3 (Remarks to the Author):

Dr Dreifuss and colleagues presented an interesting article on the estimation of the transmission dynamics of SARS-Cov-2 (that could be held as a case study for similar pathogens) in wastewater. I particularly appreciated the efforts made to investigate mathematically and computationally potential sources of bias that may impact the effectiveness of wastewater surveillance.

R3.1. However, I would like authors to strengthen their results by including additional analysis. In particular the difference in shedding is assumed to be a constant factor rescaling the incidence. This seems a rather simple assumption (considering that the shedding profile could be different due to lower days of shedding, lower viral load or a combination of both) and should be better highlighted and discussed if not revised. For instance in the stochastic simulations of variant emergence. The design of this analysis is not clear to me as it seems to be applied to the number of cases rather than to wastewater samples. If I understood correctly authors simulated two time series of cases and then to simulate undershedding they scaled down by 50% the time series of the emerging variants. This approach could also be applied (and has been applied) to simulate underreporting (ie the emerging variant being less symptomatic) and similar results have been already discussed in the context of underreporting. However it does not take into account that what is observed in wastewater surveillance is not the number of cases, but a viral load, hence missing an intermediate step. I think it would be more interesting if authors simulated the time series of cases and then from the daily number of cases generated the potentially observed values in wastewater data by exploiting a shedding profile (that could differ between variants). In this way the cumulative effects

of a case shedding for half the days or shedding half the dose due to being infected by a different variants would be correctly investigate in the frame of wastewater surveillance (that is blind to the actual number of cases).

Thank you very much for your comment. We had initially focused on differences in total shedding because we believe these are the most relevant for two key reasons that we make clearer in the revised manuscript: they are the variations most commonly observed in experimental data (see for example ref. 16), and they have the greatest impact on the relationship between incidence and loads (Equation 1). However, we agree that exploring other types of differences in shedding, such as shorter shedding times, is also important. This concern was indeed shared by the other reviewers (see for example our answers to R1.3.a, R1.5., R2.1.), and following your recommendation, we have now thoroughly explored the effects of not only varying total shedding, but also the effects of arbitrary changes in shedding shape (Figure 2, Supplementary Tables 1,2, Supplementary C, D). Concerning the stochastic simulations, we have first simulated the time series of incidence, from which we have then simulated the time series of loads in wastewater by convolving with a shedding load profile. We have made that procedure clearer in the manuscript. We appreciate your comments and suggestions, and it has very much helped make our findings more robust and general.

R3.2. Another aspect that needs revision is the assumption of a constant performance of the sewage system in maintaining viral load. This assumption is not adequately discussed in the manuscript as the whole mathematical analysis assumed that the incidence in the population is unambiguously measured in the population (by wastewater samples) while different dilutions (or other factors) could bias this estimate.

Thank you very much for your comment. We agree that wastewater transport dynamics, the performance of the sewage system, including dilution and degradation, can impact viral load estimates and are an important area of research. However, our focus here is on how differential shedding between variants affects incidence estimates, rather than external factors influencing viral transport. We do not expect wastewater fate and transport to differentially affect variants. Still, if that were to be the case, changes in loads would be akin to changes we would observe due to a difference in shedding load profiles. As such, the effect would be also covered by our results.

Reviewer 1

This is a revised manuscript that we have reviewed earlier. We have read the revised draft and will provide some comments below, focussing on presentation and the equations in the appendices. Of note, we have not checked all materials in full detail. Overall, we believe the authors have done a very good job responding to our earlier comments, and it is clear that a major effort has been invested to get the manuscript in better shape. Below we provide a point-by-point list of comments

Recommendation

Based on the above evaluation and specific comments below that are largely positive I recommend a further (minor?) revision. There are a number of inconsistencies and hidden assumptions in the Appendices, and it would be good to correct those in particular, and point how the results are affected by the underlying assumptions.

We thank the reviewers for their careful re-evaluation of our revised manuscript and for their positive feedback. We also appreciate the additional comments provided in this round of review, which helped us improve presentation and clarity of our results. We are especially grateful for the thorough and attentive reading of the Supplementary Materials, which helped identify and correct several issues. We have carefully addressed all points raised, as detailed in our point-by-point responses below.

Specific comments

p1 "... will occur close to infection." Not true as stated. Please elaborate and make this more precise (ie give an intuition from the onset).

Thank you very much for pointing this out. Our introduction previously read "loads are indicative of COVID-19 incidence rather than prevalence, as most of the shedding will occur close to infection", citing from the Hoffman et al. study. As we are here not studying indicators based on prevalence at all in the rest of the study, the question of whether wastewater loads correlate most with incidence or prevalence is beyond the scope of the paper. We have thus elected to remove this sentence.

We have now updated the sentence to explain that although prolonged low-level shedding can occur, fecal shedding typically decays exponentially with a short half-life after infection, making wastewater concentrations indicative of recent infections, i.e. indicative of incidence. Of note, we no longer state whether wastewater loads are more indicative of incidence rather than prevalence.

p2, equation (1). Please explain the equation in words and give intuition. Not all readers will have the time and ability to read through the appendices. You might also want to elaborate a bit more on wastewater data as an indicator of infection prevalence versus infection incidence here or at some other point in the Introduction.

We have revised the main text above equation (1). We now describe the equation in words to give more intuition. We now say that “the shedding profile can cause the observed viral loads $X^w(t)$ in wastewater to differ from the true incidence $X(t)$. The total amount l of viral particles shed throughout an infection will scale $X^w(t)$ with respect to $X(t)$, and the mean shedding time μ introduces a delay.”, which summarises equation (1).

Regarding prevalence versus incidence, as we have explain as a response to the previous comment, we have now entirely deleted the argument from our manuscript, as it is beyond its scope.

p3, both sections. Please give an intuition in biological terms of the results. Now I feel that many readers (including myself to a certain extent) may fail to get any intuition. This also applies to most other paragraphs in the Results section, results are simply stated without much help for the reader.

Thank you very much for this valuable suggestion. We have made an additional, different derivation of these results, which gives more biological intuition. We describe the key idea in the results, saying that “Under [the assumption of constant growth rates], the shedding load profile only affects the wastewater loads by scaling them with respect to incidence, but does not further distort the signal. As a result, the observed progression of the relative loads will appear only shifted in time with respect to the true relative incidence.” In the methods, we contextualise this with equation (6) by writing “The intuition behind these derivations is that, if the observed loads correspond to scalings of the true incidences, then the logit transformed relative loads $\phi^w(t)$ will be equal to the logit transformed relative incidence $\phi(t)$ up to an additive constant, and therefore have the same time derivative. To show that this applies to arbitrary shedding profiles in the case where transmission rates are constant, we can use a key property of exponential growth where convolution with a distribution is equivalent to a scaling.” This now also gives more intuition to the argument presented in the second section about robustness to scaling of the shedding load profiles.

We give a full description of this additional and more intuitive derivation in the supplementary (Supplementary equations 19-22).

.p10, Discussion. Here and at other places I feel that you are very much focussed on arguing that your fitness estimates are unbiased and robust. They are not always however (e.g., Figure 2), and I believe that are more balanced discussion (perhaps also extending this to the Abstract) specifying when and under which conditions the results are robust and unbiased and when they are not could further strengthen the manuscript. Put bluntly, parts of the manuscript now read as advocacy that is not entirely supported by the analyses.

We agree and have added a more nuanced discussion of the results. In particular, we now state more clearly that estimates are unbiased *under the assumptions of the selection model*.

We also write more explicitly that our results show that some types of changes in shedding could amplify inherent biases when the selection model is misspecified: “When relaxing those assumptions, estimates of selection are biased, regardless of whether they are computed from true relative incidence or from observed relative loads. Our results show that some types of changes in shedding profiles, mainly changes in mean shedding time, can amplify this bias in the wastewater-based estimates.”

Finally, we discuss more explicitly that “even under relaxed selection model assumptions, we find that differences in total shedding alone, which are the typical changes observed between variants, do not compromise the robustness of selection estimates.” Of note, the discussion also now includes a paragraph to discuss possible compounding effects of changes in generation time.

We have updated the abstract accordingly, by including the sentence: “We show that [estimates of selection advantage] are robust to differences in shedding between variants under a wide range of assumptions, and identify specific conditions under which this robustness may break down. “.

p6 ff. We did not check the materials on the new PCR.

p7, figure 1. Avoid the red-green colour combination, as 5-10% of males are red-green colourblind.

We thank the reviewer for this important observation. We have updated Figure 1 as well as Supplementary Figure 5 to use a colorblind-friendly palette that avoids red-green combinations.

p8, Figure 2. Is "baseline" explained in the legend and the text referring first to the figure?

We have clarified that baseline refers to estimates from the ground truth relative incidences in the figure legend and in the text.

Appendices

Under equation (6). $R_y \approx 1$ seems like a strong assumption. Please explain why this is needed and whether it is biologically reasonable.

We have added an explanation in the text clarifying that this is a standard assumption made in models of variant competition. We now explain that these models make this strong assumption as R_y is typically not known.

Under equation (7), it is assumed that the advantage $s = \beta_x(t)/\beta_y(t) - 1$ is constant, i.e. $\beta_x(t) = c\beta_y(t)$. Could this assumption be made clearer in the text?

We have added mention of the assumption of constant selection in the text. We have made sure now that this core assumption is mentioned in the introduction, results, methods and supplementary.

In equation (9), the convolution is defined as $(X * g)(t) = \int_{\tau} X(\tau)g(t - \tau)d\tau$, whereas the equivalent definition of $(X * g)(t) = \int_{\tau} X(t - \tau)g(\tau)d\tau$ is used throughout the appendix. It would be more consistent to use the latter definition.

Thank you very much for spotting this. We have now changed the notation as requested to make it more consistent.

In equation (10), $X(t - \tau)$ is approximated with a Taylor approximation as $X(t - \tau) \approx X(t) + X'(t)(t - \tau - t) + \frac{X''(t)(t - \tau - t)^2}{2} = X(t) - X'(t)\tau + \frac{X''\tau^2}{2}$, whereas the text has a minus-sign in front of the second derivative. This should be changed in a plus sign throughout.

We thank the reviewer warmly for catching this sign error. We have now corrected it.

C.iii. In the text it is stated “We have shown above that when this assumption does not hold, (...)”, where does this refer to?

We have now added a direct reference to Supplementary Equation 7, which improves clarity.

In equation (24), it is implicitly used that the advantage s is assumed to be constant (see also under equation (7)). Please make this assumption explicit before the derivation of equation (24).

We have now made this assumption more explicit above the derivation of Supplementary Equation 28 (formerly 24). We also directly reference Supplementary Equation 7 now.

C.iv. Where does the last sentence “Following the same argument as above, these estimates will be robust to changes in total shedding for variant X ” refer to?

We have added a reference pointing to Supplementary C (ii).

C.vi. Again, the derivations are implicit on the fact that s is constant, make this explicit.

In equation (34), do you want to compare $s^{\hat{w}_2}(t)$ with $s^{\hat{w}}(t)$ or with $s^{\hat{t}}(t)$?

We have added mention of the assumption of constant selection here. We also make sure that Supplementary Equation 38 (formerly 34) correctly displays the comparison between $s^{\hat{w}_2}(t)$ and $s^{\hat{w}}(t)$.

There should be a minus sign in front of the middle and right part of equation (34).

Thank you very much for catching this. We have now fixed the definition of $\Delta\mu$ from $\Delta\mu = \mu_x - \mu_y$ to $\Delta\mu = \mu_y - \mu_x$ here, which makes it more consistent with the rest of the text and fixes this sign error.

C.vii. In equation (36) the text “and” is in italics.

Thank you very much, we have changed this formatting.

C.viii. In the text it says “In the case where we additionally have non-constant generation times, (...)” how does “additionally” fit here as β_x is no longer time-dependent.

We have rewritten such that it is clear now that the generation times are different between variants and are additionally time-varying.

Under equation (42) γ^{-1} is referred to as the generation time, while both variants have a different generation. Could you clarify which generation time this is.

Thank you very much for pointing this out, we have now changed γ^{-1} to γ_c^{-1} in Supplementary Equation 46 and 47 (formerly 42 and 43). We are describing now that “ γ_c^{-1} is the (incorrectly assumed) constant generation time.”

In equation (43), the equality sign should be an approximate equal to sign.

Thank you very much for pointing this out, we have corrected this in Supplementary Equation 47 (formerly 43).

D.ii. Make explicit that the constant c should be larger than 1 for the estimations to hold.

Thank you for this comment. We have made clear in the text now what are the restrictions on c .

Under equation (49), ϕ' is made equal to $s\gamma$, which only holds if $R \approx 1$. As the reproduction number is estimated here, it would be nice to again make this assumption here explicit. Please also specify the biological conditions under this is reasonable.

We have made the assumption explicit under equation 53 (formerly 49). We clarified that we make this assumption here to obtain a further simplification of the approximate bound. Specifically, the explanation now reads: “Simplifying by assuming $R_y \approx 1$ (which holds when the variant Y is in the endemic regime), this yields the bound ...”.

D.iii. In equation (51), we find the use of I in the right hand part of the equation confusing and would prefer $X(t)$, and a reference to supplementary equation 11 better suited than a reference to equation 1.

Thank you very much for pointing this out. We have indeed replaced I with X to avoid confusion. We have replaced the reference to equation 1 with a reference to equation 11, which indeed fits better here.

In equation (52), the final approximate equal to-sign can be changed in equality.

Thanks a lot for catching this. We have fixed Supplementary Equation 56 (formerly 52) to include an equal sign instead of the approximate equal-to sign.

For the estimations in equations (53) and (54) to hold, β_x is assumed larger than γ , which should be made explicit.

We have made the clearer the restriction on $(\beta_x - \gamma)\Delta_\mu f(t)$ below equation 57 (formerly 54) to avoid division by zero.

Dispersion of the generation interval time. How do the reproduction numbers of 0.6 and 2.2 compare to the assumption of $R_y \approx 1$?

We have clarified that we indeed used the assumption $R_y \approx 1$ for estimating the selection advantage in both scenarios. In that case and as we note, the estimates of selection advantage will be biased whether estimated from true relative incidence or wastewater loads. What we observe from the simulations is that, selection estimates from the scenario with different shedding are not more biased than the ones from the scenario without differences in shedding, i.e. the estimator is robust to changes in total shedding.

Reviewer 2

Thank you for the opportunity to review the revised manuscript. My main comments are:

1. My previous comment R2.2 was not adequately addressed. While the manuscript examined the impacts of viral shedding profiles and generation time on wastewater-derived epidemiological estimates, it largely treated these as independent factors. The paper would be strengthened by explicitly discussing and potentially modeling the intrinsic biological link between shedding patterns (timing and magnitude) and an individual's infectiousness profile, as this profile dictates generation time and likely correlates with shedding.

Thank you very much for your comment. We are now more explicitly modeling variants which can have correlated differences both in shedding profiles (shape or total shedding) and in mean generation times, and we have added a full paragraph in the discussion. We discuss how this could arise due to differences in the internal viral load affecting both shedding and transmission dynamics (infectiousness profiles). We

give the example in the supplementary C (vii) and in the discussion of the hypothetical case of a variant with altered within-host kinetics leading to earlier or more concentrated transmission and shedding.

Our derivations, which we clarify in the methods and in the supplementary C (vii), show that failing to account for differences in mean generation time introduces bias in estimates of selection, whether computed from true relative incidence or from observed relative loads (in wastewater). However, in the case of differences in mean generation time, arbitrary simultaneous changes in shedding profiles do not amplify the bias (with respect to a direct case-observation based measure). We have also clarified this in the results.

We did not explicitly model nor simulate scenarios where both the shape of the shedding profile and the shape of the generation time distribution are simultaneously changed for variants with different transmission rates. This would get arbitrarily complex and is beyond the scope of this study. However, in the discussion we now note that we expect that in such situations the effects of bias could indeed compound.

2. The analysis tested sensitivity to generation time dispersion but should have further explored how altered shedding profiles, if mechanistically linked to infectiousness, could systematically change the generation time distribution (both mean and shape). It is unclear if the current robustness checks for generation time fully capture these coupled dynamics.

Thank you very much for your comment. As discussed in the answer to (1.), we have now more explicitly modeled variants which have correlated differences both in shedding profiles (shape or total shedding) and in mean generation times. We describe more clearly the effect of these coupled dynamics.

Our simulations described in Supplementary D (iv), Supplementary Figure 5 and summarized in the Methods and Results section explore the coupling of a change in the shape of the generation time distribution with a change in the total shedding, and find that wastewater based estimates of selection are not more biased than estimates based on true relative incidence.

As mentioned in the answer to (1.), we did not simulate scenarios where a change in the shape of the generation time distribution is coupled with a change in the shape of the shedding load distribution for variants with unequal transmission rates, as this is beyond the scope of this study. We however briefly mention it in the Discussion.

3. The conclusion that estimates of selection advantage are robust to separate differences in total shedding and generation times warrants further discussion. The authors should address the implications if shedding characteristics and infectiousness profiles (and thus generation times) are coupled—for example, if a variant exhibits a different shedding profile due to an altered infectiousness course. The potential impact of such biologically plausible couplings on the robustness of the findings needs more thorough consideration.

Thank you very much for your comment. As mentioned in the answers to (1.) and (2.), we now explicitly discuss biologically plausible couplings between shedding and infectiousness profiles. As described in the Discussion and Supplementary C (vii), we explicitly model variants with correlated changes in shedding and mean generation time. We discuss biologically possible couplings reflecting altered within-host kinetics. We also discuss scenarios not covered by our results where wastewater-based estimates of selection could lose their robustness.

Response to reviewers:

Reviewer #1 (Remarks to the Author):

I am happy with the latest set of revisions.

We warmly thank the reviewer for their helpful feedback on the manuscript during the reviewing process.

Reviewer #2 (Remarks to the Author):

Thank you for inviting me to review the revised manuscript again. The manuscript improved significantly after explicitly accounting for individual shedding in the wastewater-derived epidemiological estimates. I concur with Reviewer 1's comment that the manuscript would benefit from a more balanced discussion regarding the robustness of the fitness estimates.

We thank the reviewer for their positive feedback on our manuscript. We have indeed in the last revision round added a more nuanced discussion of the results. In particular, we state more clearly that estimates are unbiased under the assumptions of the selection model.

We also write more explicitly that our results show that some types of changes in shedding could amplify inherent biases when the selection model is misspecified: "When relaxing those assumptions, estimates of selection are biased, regardless of whether they are computed from true relative incidence or from observed relative loads. Our results show that some types of changes in shedding profiles, mainly changes in mean shedding time, can amplify this bias in the wastewater-based estimates."

Finally, we discuss more explicitly that "even under relaxed selection model assumptions, we find that differences in total shedding alone, which are the typical changes observed between variants, do not compromise the robustness of selection estimates." Of note, the discussion also now includes a paragraph to discuss possible compounding effects of changes in generation time.

We have updated the abstract accordingly, by including the sentence: "We show that [estimates of selection advantage] are robust to differences in shedding between variants under a wide range of assumptions, and identify specific conditions under which this robustness may break down. "

With caution, the statement "Another standard assumption in evolutionary models used to estimate the fitness advantages of SARS-CoV-2 variants is that variants have the same generation time" should not be presented as a "standard" assumption. Therefore, the subsequent claim (and throughout the manuscript), "Here, we have shown that under standard assumptions of selection models, estimates of this parameter based on wastewater-derived data are unbiased as well as robust to arbitrary changes in shedding between variants," is less

convincing. I suggest that the authors refer to this assumption as the "baseline" instead of claiming it as a "standard assumption."

We thank the reviewer for their comment. We have now edited the wording and describe the assumption as a "common" assumption across the whole manuscript. In particular, we now write: "Here, we have shown that under common assumptions of selection models, estimates of this parameter based on wastewater-derived data are unbiased as well as robust to arbitrary changes in shedding between variants,".

Editorial Note: Reviewer #1 Version 1 Review Attachment:

Review of the revised manuscript 'Estimated transmission dynamics of SARS-CoV-2 variants from wastewater are unbiased and robust to differential shedding' by David Dreifuss and colleagues, submitted to Nature Communications.

Evaluation

This is a revised manuscript that we have reviewed earlier. We have read the revised draft and will provide some comments below, focussing on presentation and the equations in the appendices. Of note, we have not checked all materials in full detail. Overall, we believe the authors have done a very good job responding to our earlier comments, and it is clear that a major effort has been invested to get the manuscript in better shape. Below we provide a point-by-point list of comments

Recommendation

Based on the above evaluation and specific comments below that are largely positive I recommend a further (minor?) revision. There are a number of inconsistencies and hidden assumptions in the Appendices, and it would be good to correct those in particular, and point how the results are affected by the underlying assumptions.

Specific comments

p1 "... will occur close to infection." Not true as stated. Please elaborate and make this more precise (ie give an intuition from the onset).

p2, equation (1). Please explain the equation in words and give intuition. Not all readers will have the time and ability to read through the appendices. You might also want to elaborate a bit more on wastewater data as an indicator of infection prevalence versus infection incidence here or at some other point in the Introduction.

p3, both sections. Please give an intuition in biological terms of the results. Now I feel that many readers (including myself to a certain extent) may fail to get any intuition. This also applies to most other paragraphs in the Results section, results are simply stated without much help for the reader.

p10, Discussion. Here and at other places I feel that you are very much focussed on arguing that your fitness estimates are unbiased and robust. They are not always however (e.g., Figure 2), and I believe that a more balanced discussion (perhaps also extending this to the Abstract) specifying when and under which conditions the results are robust and unbiased and when they are not could further strengthen the manuscript. Put bluntly, parts of the manuscript now read as advocacy that is not entirely supported by the analyses.

p6 ff. We did not check the materials on the new PCR.

p7, figure 1. Avoid the red-green colour combination, as 5-10% of males are red-green colourblind.

p8, Figure 2. Is "baseline" explained in the legend and the text referring first to the figure?

Appendices

Under equation (6). $R_y \approx 1$ seems like a strong assumption. Please explain why this is needed and whether it is biologically reasonable.

Under equation (7), it is assumed that the advantage $s = \frac{\beta_x(t)}{\beta_y(t)} - 1$ is constant, i.e. $\beta_x(t) = c \times \beta_y(t)$. Could this assumption be made clearer in the text?

In equation (9), the convolution is defined as $\int_{\tau} X(\tau)g(t - \tau)d\tau$, whereas the equivalent definition of $\int_{\tau} X(t - \tau)g(\tau)d\tau$ is used throughout the appendix. It would be more consistent to use the latter definition.

In equation (10), $X(t - \tau)$ is approximated with a Taylor approximation as $X(t - \tau) \approx X(t) + X'(t)(t - \tau - t) + \frac{X''(t)(t - \tau - t)^2}{2} = X(t) - X'(t)\tau + \frac{X''(t)\tau^2}{2}$, whereas the text has a minus-sign in front of the second derivative. This should be changed in a plus sign throughout.

C.iii. In the text it is stated "We have shown above that when this assumption does not hold, (...)", where does this refer to?

In equation (24), it is implicitly used that the advantage s is assumed to be constant (see also under equation (7)). Please make this assumption explicit before the derivation of equation (24).

C.iv. Where does the last sentence "Following the same argument as above, these estimates will be robust to changes in total shedding for variant X" refer to?

C.vi. Again, the derivations are implicit on the fact that s is constant, make this explicit.

In equation (34), do you want to compare $s^{\widehat{w}_2}(t)$ with $s^{\widehat{w}(t)}$ or with $\hat{s}(t)$? There should be a minus sign in front of the middle and right part of equation (34).

C.vii. In equation (36) the text "and" is in italics.

C.viii. In the text it says “In the case where we additionally have non-constant generation times, (...)” how does “additionally fit here as β_x is no longer time-dependent.

Under equation (42) γ^{-1} is referred to as the generation time, while both variants have a different generation. Could you clarify which generation time this is.

In equation (43), the equality sign should be an approximate equal to sign.

D.ii. Make explicit that the constant c should be larger than 1 for the estimations to hold.

Under equation (49), ϕ' is made equal to $s\gamma$, which only holds if $R_y \approx 1$. As the reproduction number is estimated here, it would be nice to again make this assumption here explicit. Please also specify the biological conditions under this is reasonable.

D.iii. In equation (51), we find the use of I in the right hand part of the equation confusing and would prefer $X(t)$, and a reference to supplementary equation 11 better suited than a reference to equation 1.

In equation (52), the final approximate equal to-sign can be changed in equality.

For the estimations in equations (53) and (54) to hold, β_x is assumed larger than γ , which should be made explicit.

Dispersion of the generation interval time. How do the reproduction numbers of 0.6 and 2.2 compare to the assumption of $R_y \approx 1$?